# Insights into the Cell Division of *Neospora caninum*

**DOI:** 10.3390/microorganisms12010061

**Published:** 2023-12-28

**Authors:** Ramiro Tomasina, Fabiana C. González, Soledad Echeverría, Andrés Cabrera, Carlos Robello

**Affiliations:** 1Laboratorio de Interacciones Hospedero Patógeno, Institut Pasteur de Montevideo, Montevideo 11400, Uruguay; rtomasina@pasteur.edu.uy (R.T.); fcgonzalez@pasteur.edu.uy (F.C.G.); echeverria@pasteur.edu.uy (S.E.); cabrera@pasteur.edu.uy (A.C.); 2Departamento de Parasitología y Micología, Facultad de Medicina, Universidad de la República, Montevideo 11800, Uruguay; 3Departamento de Bioquímica, Facultad de Medicina, Universidad de la República, Montevideo 11800, Uruguay

**Keywords:** apicomplexa, endodyogeny, *Neospora caninum*

## Abstract

*Neospora caninum* is an apicomplexan protozoan parasite responsible for causing neosporosis in a range of animal species. It results in substantial economic losses in the livestock industry and poses significant health risks to companion and wild animals. Central to its survival and pathogenicity is the process of cell division, which remains poorly understood in this parasite. In this study, we explored the cell division of *Neospora caninum* using a combination of modern and classic imaging tools, emphasizing its pivotal role in perpetuating the parasite’s life cycle and contributing to its ability to persist within host organisms. We described the intricacies of endodyogeny in *Neospora caninum*, detailing the dynamics of the cell assembly and the nuclear division by ultrastructure expansion microscopy and regular confocal microscopy. Furthermore, we explored the centrosome dynamics, the centrioles and the apicoplast through the advancement of the cell cycle. Our analysis described with unprecedented detail, the endodyogeny in this parasite. By advancing our understanding of these molecular mechanisms, we aimed to inspire innovative strategies for disease management and control, with the ultimate goal of mitigating the devastating impact of neosporosis on animal health and welfare.

## 1. Introduction

Apicomplexan parasites are a group of more than 6000 species of intracellular organisms with complex life cycles involving intricate interactions with their hosts [1]. Among these parasites stands *Neospora caninum*, responsible for causing neosporosis, which is a significant veterinary concern in both domestic and wild animals [2]. The species *N. caninum* belongs to the phylum Apicomplexa, class Sporozoans, subclass Coccidia and genus Neospora. Like many apicomplexans, this obligate intracellular parasite exhibits remarkable adaptations that enable it to proliferate and persist within its host, posing challenges for disease control and treatment [3].

The life cycle of *N. caninum* unfolds within a complex web of host interactions, including definitive and intermediate hosts, where the parasite undergoes a series of intriguing developmental transformations [2,4,5]. Depending upon the specific host they invade, the parasite undergoes either asexual or sexual reproduction. Sexual reproduction occurs in the intestine of canids, the definitive hosts [5]. However, little is known about the replication schemes that *N. caninum* needs to follow to produce the oocyst found in the feces of this host. Asexual reproduction occurs in the intermediate hosts, including cattle [3]. During the asexual stage, the parasite undergoes endodyogeny. This is an intricate process that involves assembling two daughter cells inside a mother cell as the nucleus replicates and segregates in each newly formed cell [1,6]. Previous works using transmission electron microscopy (TEM) have shown evidence of this parasite following this scheme of replication [4,7,8]. By TEM, the morphology of *N. caninum* looks similar to the morphology observed in *Toxoplasma gondii*, which is coherent with the fact that they are highly closely related parasites [4,7,8]. *N. caninum* has one mitochondrion, an apicoplast, rhoptries, micronemes and two main microtubule organizing centers (MTOCs). The two MTOCs of this parasite are the apical polar ring and the centrosome [4,7,8]. 

As an apicomplexan parasite, *N. caninum* pathogenicity relies on its replicative efficiency [1]. The process of cell division enables the parasite to multiply and disseminate throughout the host organism. In many apicomplexan parasites, centrosomes, as the main microtubule organizing centers of the cells, are implicated as pivotal orchestrators of cell division [9,10,11,12,13]. While the architectural diversity of centrosomes among those apicomplexan parasites is wide, a unifying principle seems to underscore their role in coordinating nuclear segregation and the structural assembly of daughter cell scaffolds [10,11,12,13,14]. In *T. gondii*, a closely related parasitic counterpart to *N. caninum*, the critical role of the centrosome in regulating the endodyogeny process has been documented [9,10,11,15]. The centrosome in *T. gondii* exhibits a tripartite organization, consisting of outer, middle, and inner cores, delineated by their relative position to the nucleus [9,10,11,15]. Each core encompasses distinct functional domains [9,10,11,15]. The outer core houses the centrioles [9] and is implicated in the budding process [10,11], the inner core in nuclear segregation [9], and the middle core in maintaining overall cohesion [15]. The intricate composition of the centrosome in this parasite aligns with the intricate nature of the processes orchestrated by this organelle during the cell division of the parasite. The centrosome keeps a semi-closed nuclear mitosis and an internal cell-building process linked spatially and temporally as the endodyogeny in this parasite takes place [9,10,11,15].

In *Plasmodium* spp. many studies have shown the role of the centrosome (known as the centriolar plaque) in the coordination of nuclear segregation but with the difference of having a centrosome lacking centrioles [12,13,16,17]. Remarkably, little is known about the centrosome of *N. caninum*.

Understanding the mechanisms governing cell division in this parasitic protozoan is paramount in shedding light on its pathogenesis, host interactions and ultimately, the development of effective therapeutic interventions. With its high affinity for the placental and neural tissues, *N. caninum* can perpetuate within the host, contributing to vertical transmission leading to congenital infection in susceptible animals [2,18]. 

By describing the molecular mechanisms of cell division in *N. caninum*, this study endeavors to contribute to a broader understanding of parasitic pathogenesis. Moreover, a comprehensive analysis of the cell division processes could help in unravelling the parasite’s ability to adapt and persist in diverse host environments, providing critical insights into its epidemiology and transmission dynamics.

## 2. Materials and Methods

### 2.1. Cell Culture

Liverpool’s *N. caninum* tachyzoites were maintained in Vero cells, grown at 37 °C with 5% CO_2_ in bicarbonate-buffered Dulbecco’s modified Eagle’s medium (Gibco^TM^, Thermo Fisher Scientific, Waltham, MA, USA), supplemented with 10% inactivated bovine serum (Gibco^TM^, Thermo Fisher Scientific, Waltham, MA, USA) and penicillin/streptomycin (Invitrogen, Carlsbad, CA, USA) [19].

### 2.2. Optical Microscopy

Vero cell line cells were grown on coverslips until 85–90% confluence was achieved. They were then washed with PBS and inoculated with 1 × 10^6^ parasites. Then, 24 h later, the coverslips containing the cells infected with parasites were washed with PBS and fixed with methanol at 4 °C for 5 min. Subsequently, they were blocked with 5% BSA in phosphate-buffered saline (PBS). The following primary antibodies were used at a 1:1000 dilution: mouse anti-Centrin-1 (centrosome marker) (Cell Signaling, Frankfurt, Germany, 04-1624), rabbit anti-Cpn60 (apicoplast marker) [20] (kindly provided by Dr. Sébastien Besteiro) and mouse anti-acetylated tubulin (acetylated tubulin structure-marker) (Sigma, Saint Louise, MO, USA, T7451). Secondary antibodies were as follows: goat anti-mouse Alexa Fluor 488 (Invitrogen, Rockford, IL, USA, A28175), goat anti-mouse Alexa Fluor 594 (Invitrogen, Rockford, IL, USA) and anti-rabbit Alexa Fluor 594 (Invitrogen, Rockford, IL, USA). All antibodies were incubated at a 1:1000 dilution, followed by 5 min of incubation with 1 μg/mL DAPI in PBS. The coverslips were mounted with ProLong Gold Antifade Mountant (Invitrogen, Rockford, IL, USA, P36930) of Fluoroshield with DAPI (Sigma, Saint Louise, MO, USA, F6057).

### 2.3. Ultrastructure Expansion Microscopy

Ultrastructure expansion microscopy (UExM) was executed following the previously outlined methodology for *T. gondii* [21], maintaining fidelity to the original protocol. In summary, coverslips containing cells harboring parasitic infections were subjected to a 5 h incubation period with a blend of 0.7% acrylamide (AA) and 1% formaldehyde (FA) at 37 °C. Subsequently, the gelification process was initiated by introducing a solution of 19% sodium acrylate (SA), 10% AA and 0.1% BIS-AA in PBS. This catalytic process occurred over 1 h at 37 °C. The subsequent phase encompassed the denaturation of proteins through a 1.5 h incubation at 95 °C, thereby facilitating the expansion of the gel that encapsulated the parasites. This expanded gel was immersed in water for overnight expansion.

To visualize the ultrastructure of the parasites and discern the spatial distribution of specific proteins, a conventional indirect immunofluorescence protocol was applied, as described previously for *T. gondii* [21]. Mouse anti-acetylated tubulin (acetylated tubulin structure-marker) (Sigma T7451) and goat anti-mouse Alexa Fluor 594 (Invitrogen) were both used in a dilution of 1:500 in PBS. A Zeiss LSM880 confocal microscope (Oberkochem, Germany) with a Plan-Apochromat 63×/1.40 oil immersion objective was used, and post-image acquisition and processing were performed with ImageJ v1.54g (NIH) and Zeiss ZEN blue edition v2.0 software. The 3D model in Figures 1 and 4 was built using Agave v1.5.0 free software [22]. 

## 3. Results

### 3.1. A 3D Model for N. caninum Vacuoles Using UExM 

During asexual division, *N. caninum* invades the host cells and replicates following endodyogeny, a biological process characterized by the intracellular assembly of two daughter cells within a maternal cellular entity. This is concomitant with the duplication and orderly partitioning of the nucleus into each nascent progeny cell [8]. Previous studies have allowed the observation of this process by TEM [4,7,8], but none of these have been focused on the dynamics of the cell cycle of this parasite. Using ultrastructure expansion microscopy (UExM), a new technique previously used for other apicomplexan parasites [9,14,16,21,23,24], we were able to observe the complexity of the vacuoles of the parasites, observe endodyogeny and create a 3D [22] computational model, as shown in Figure 1. In contrast to what is seen in *T. gondii*, where the vacuoles of parasites tend to maintain an organized rosette structure as their numbers increase [25,26], the vacuoles of *N. caninum* appear more disorganized as they enlarge, as illustrated in Figure 1 and in Appendix A. Remarkably, the spatial distribution of the parasites involves both orthogonal and parallel arrangements, forming large and unorganized vacuoles as they grow in number.

### 3.2. The Centrosome of N. caninum

Centrosomes are the main microtubule organizing centers of the cells (MTOCs), playing essential roles in the coordination of cell division in many eukaryotes. Apicomplexans are not the exception to this. In *T. gondii*, the centrosome is crucial in coordinating cell division, acting as a linker between the buddying process and nuclear mitosis [6,9,10,15]. In *Plasmodium* spp., the centriolar plaque (a structure similar to the centrosome but lacking centrioles) is important in coordinating nuclear mitosis [27]. 

Due to the phylogenetic proximity between *T. gondii* and *N. caninum*, it is not implausible to hypothesize that the centrosome in *N. caninum* is playing a similar role. To address this hypothesis, we performed bioinformatics searches for centrosomal genes found in *T. gondii* and *Plasmodium spp.* Those searches conclude that in *N. caninum*’s genome, most genes assigned to the centrosome in *T. gondii* [14] are present (Appendix A). This finding indicated that the composition of the centrosome of these parasites should be similar. The next question we addressed was whether the centrosome of this parasite had a cell cycle pattern linked to the progression of endodyogeny, as occurs in other apicomplexan parasites (i.e., in *T. gondii* the centrosome undergoes duplication once every cell cycle and only during mitosis [1,9,10]). If *N. caninum* behaves similarly to *T. gondii*, we expected to observe two scenarios concerning the centrosome—unduplicated and duplicated [1,9,10]. To probe this hypothesis, we used anti-Centrin-1 antibodies to observe the position of the centrosome in an unsynchronized population of parasites. 

As is depicted in Figure 2, both expected scenarios occur. 

### 3.3. Centriole Dynamics through the Endodyogeny in N. caninum

Due to inherent limitations in optical microscopy resolution, akin to observations made of other apicomplexan parasites like *T. gondii* [9,10,11,15], visualization of the centrioles within *N. caninum* was not possible using confocal microscopy. To overcome this constraint, we employed UExM for the first time in this parasite, achieving a fourfold expansion factor, thereby enabling discernment of the centrioles housed within the centrosome. Utilizing an anti-acetylated tubulin antibody in conjunction with UExM, we could observe the dynamic comportment of centrioles and the assembly of the mitotic spindle in *N. caninum*. During endodyogeny, our observations revealed a sequence wherein centrioles undergo duplication and strategic repositioning, thereby facilitating the meticulous assembly of scaffolds within the ensuing daughter cells. During interphase, two centrioles are exhibited with a parallel orientation, reminiscent of the patterns observed in *T. gondii*, as depicted in panel I of Figure 3. Upon the initiation of endodyogeny, centriolar duplication ensued, culminating in the emergence of four centrioles in a mutually facing arrangement. Simultaneously, the assembly of the mitotic spindle unfolded, synchronized with this centriolar duplication, as illustrated visually in panel II of Figure 3. As the endodyogeny process advanced, the daughter cells underwent an incremental size increase, and the gradual separation of centrioles occurred in tandem with the eventual disjunction of the mitotic spindle. The culmination of this intricately orchestrated process materialized in the formation of two fully assembled daughter cells, housed within the confines of the mother cell, as depicted in panel IV of Figure 3. 

### 3.4. Apical Microtubule Organizing Center of the Cell in Neospora caninum

*N. caninum* exhibits two microtubule organizing centers (MTOCs), a characteristic shared with numerous apicomplexan parasites [7,8]. One of these MTOCs appears as an apical ring situated at the apical region of the parasite, while the other corresponds to the centrosome. This apical ring, commonly called the apical polar ring in other apicomplexans, plays a pivotal role in orchestrating the arrangement of the subpellicular microtubule scaffold [7,8]. Intriguingly, through the application of ultrastructure expansion microscopy (UExM) and AGAVE v.1.5.0 3D software [22], we were able to discern and show the 22 subpellicular microtubules that constitute the scaffold responsible for shaping zoites in *Neospora caninum*, a phenomenon that has previously been suggested [7] but has hitherto been unobservable through immunofluorescence microscopy (Figure 4B). This observation was confirmed consistently across all the tachyzoites subjected to detailed analysis (Appendix A). A global view is shown in Appendix A.

### 3.5. Apicoplast Dynamics during Endodyogeny

During endodyogeny in apicomplexan parasites, many organelles are replicated and segregated into the daughter cells [6,28]. Among these organelles is the apicoplast, a crucial organelle for the apicomplexan parasites’ survival [29,30,31,32].

Using the antibody anti-Cpn60 [20], which recognizes the apicoplast in *T. gondii* and *Plasmodium* spp. [32,33], we could follow this endosymbiont through the cell cycle by confocal microscopy, as shown in Figure 5. Confocal microscopy enabled us to observe apicoplast division events, leading to the formation of a large apicoplast that is divided and distributed equitably to each daughter cell. Moreover, we noted a distinct change in apicoplast shape, transitioning from a condensed and elongated form during the early stages of division to a condensed configuration towards the final stages, similar to that previously observed in *T. gondii*. We then asked whether the apicoplast and the centrosome could follow a cell cycle pattern, as previously described for *T. gondii* [30].

According to what has been observed in *T. gondii* [32], as the cell cycle advanced, the apicoplast changed its position towards the centrosome but was always linked to it (Figure 5b). In the same way, we could observe all the stages originally defined for *T. gondii* [32]. The apicoplast displayed a dynamic localization relative to the centrosome throughout the cell cycle. During the G1 phase, when the centrosome had not yet been duplicated, the apicoplast was located between the centrosome and the apical portion of the parasite. The segregation of centrosomes drove the segregation of the duplicated apicoplast during the cell division phases. As time advanced, this was accompanied by a change in position, as described earlier, to a basal position. It localized between the centrosome and the nucleus during the onset of nuclear segregation and remained in this position until the daughter cells assembled.

## 4. Discussion

Apicomplexan parasites rely on their incredible capacity to invade and replicate inside the host cells [1]. These parasites share a distinctive structural feature known as the apical complex, from which they derive their name. The apical complex is a pivotal invasion structure, encompassing secretory organelles such as rhoptries and micronemes. Within the apical complex, additional structures are discernible, including the conoid, a tubulin-based structure, and the apical polar ring (APR), which functions as the microtubule organizing center (MTOC) for the subpellicular microtubules. This arrangement imparts shape to the parasites [34]. Once inside the host cells, the parasites start to replicate. Plasticity rules these parasites’ capacity to divide. Depending on the host they invade, apicomplexans can follow many different schemes of replication [1,6]. These intracellular parasites can follow schizogony, endopolygeny or endodyogeny [1,6]. Any of these methods of replication mean a dialogue between the created daughter cells and the nuclear mitosis [9,11]. Much of what is known about these fascinating species has been observed mainly in *T. gondii*, *Plasmodium* spp. and recently, in *Cryptosporidum* spp. [35]. However, apicomplexan parasites involve more than 6000 species, and similar does not strictly mean the same. In this study, we could describe some of the processes occurring during endodyogeny in this parasite, focusing on the apicoplast and centrosome dynamics in *N. caninum*.

From our observations, the centrosome of *N. caninum* follows a cell cycle dynamic encompassing endodyogeny. According to this observation, it is not implausible to think that the centrosome plays a role in the coordination of cell division, similar to that previously observed in *T. gondii* [9,10,11]. However, further experiments are needed to prove this hypothesis.

Using bioinformatics tools, we could observe that the main centrosomal genes reported for *T. gondii* and *Plasmodium* spp. seem to be conserved. We could also observe that *N. caninum* encodes for all the genes related to the centrosome in *T. gondii*. It would be interesting to explore whether the centrosome of this parasite keeps the three-domain centrosome spatial structure or not [9,10,15], taking into consideration that it maintains homology to the central genes of *T. gondii*, recognized for each domain [9,10,15].

In this study, we used ultrastructure expansion microscopy (UExM) for the first time in *N. caninum*. This technique allowed us to reach a factor of four times the resolution of confocal microscopy using the same microscope. UExM allowed us to determine that the centrioles contained in the centrosome are two barrels of tubulin disposed of in parallel orientation. However, due to the limitation of resolution of the microscopy techniques employed, the geometry of the barrels of microtubules forming the centrioles still needs to be explored. We could also describe the centriole dynamics through the cell cycle of this parasite and how this dynamic is linked to the mitotic spindle assembly and the daughter cell scaffolds built. We could observe the APR identifying the 22 subpellicular microtubules that built the scaffold that shapes zoites in this parasite, as previously described [7], but never observed by an immunofluorescence microscopy technique.

We could document the apicoplast cell cycle dynamic, observing how this endosymbiont changes its position as the centrosome follows its path through the endodyogeny. Remarkably, we observed a similar pattern to that previously shown for *T. gondii* in another study [32], where the centrosome and the apicoplast are always linked. Using UExM and computational software, we could create a 3D model of the vacuole of these parasites. We could observe the way tachyzoites organize inside the parasitophorous vacuole. Remarkably, *N. caninum* seems to build disorganized vacuoles, with different parasite orientations as they grow in number. Although cell division occurs almost identically to that observed for *T. gondii*, the vacuoles assembled by these two parasites as they grow in number seem to be organized differently. The mechanism regulating the number and the orientation of the tachyzoites inside the vacuoles does not appear to be related to the endodyogeny process.

## Figures and Tables

**Figure 1 microorganisms-12-00061-f001:**
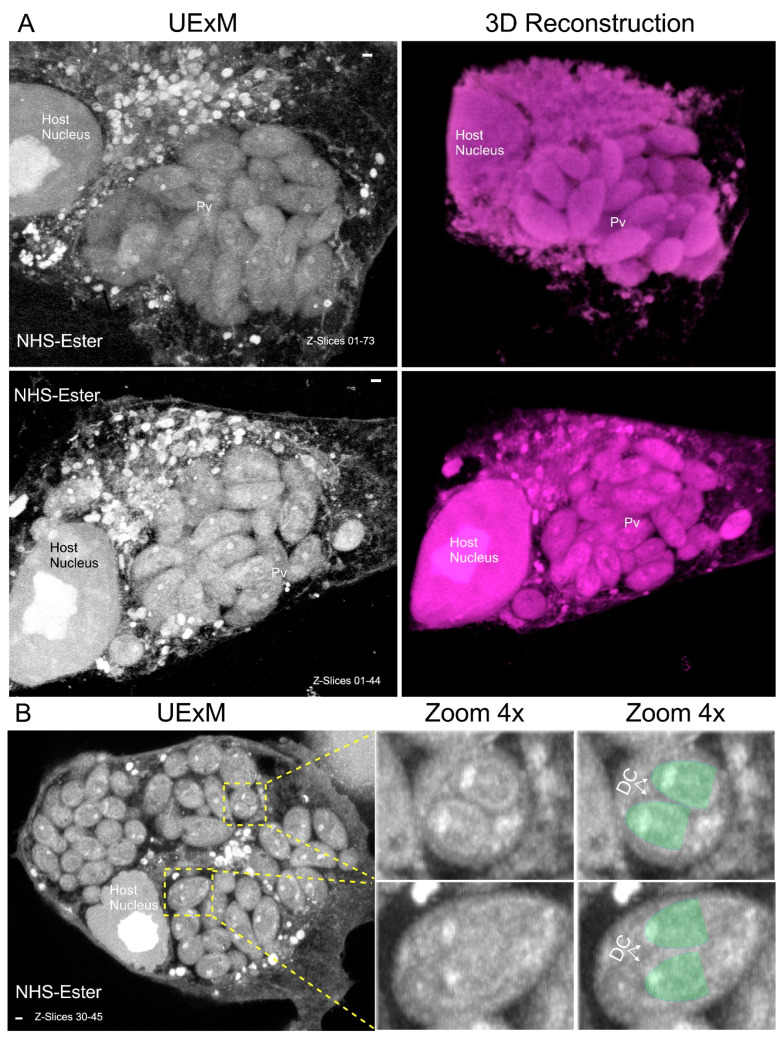
Ultrastructure expansion microscopy reveals the complexity of *Neospora caninum* vacuoles. Parasites underwent ultrastructure expansion and were subsequently treated with NHS ester. (**A**) Image depicting maximum-intensity projection z-stacks that encompass the entire parasitophorous vacuole was utilized for the generation of a three-dimensional (3D) model using Agave v1.5.0 software [22]. It is important to observe the intricate nature of the vacuole structure; “Pv” stands for parasitophorous vacuole. (**B**) Image displaying maximum-intensity projection z-stacks covering selected z-slices; it can be observed in parasites undergoing endodyogeny. DC stands for daughter cell. To facilitate observation of parasite division, DCs are highlighted in green. The scale bar consistently represents a length of 1 µm for all presented images.

**Figure 2 microorganisms-12-00061-f002:**
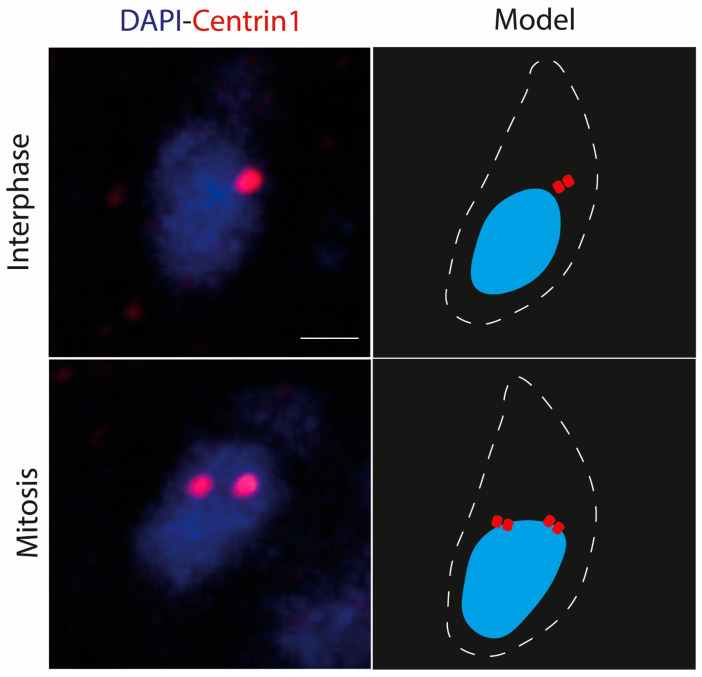
The centrosome replicates as the parasite starts cellular division. Immunofluorescence indirect assay (IFA) of *Neospora caninum* parasites stained with anti-Centrin-1 antibody (red) and DAPI (blue). Note that during interphase, one foci of Centrin 1 is observed. As the cell cycle advances, two foci of Centrin-1 are observed. White dashed lines show outline of the parasite. The scale bar is 1 µm for all images.

**Figure 3 microorganisms-12-00061-f003:**
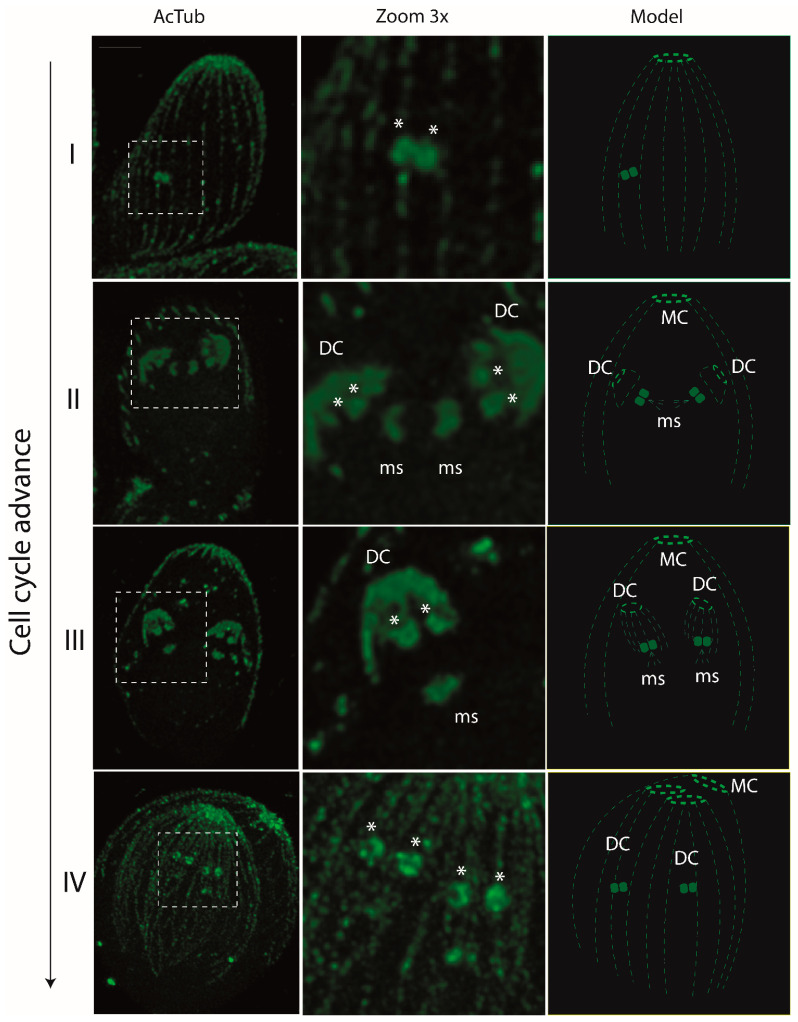
Ultrastructure expansion microscopy reveals the dynamics of the centrioles during the cell cycle of *Neospora caninum*. Parasites underwent UExM and were subsequently treated with anti-acetylated tubulin antibody. As previously observed for *T. gondii*, this protein is present in the scaffold of mother and daughter cells (MC and DC, respectively) and in the mitotic spindle (ms) and centrioles (*) [9,21]. Before mitosis begins (panel I), two centrioles disposed in a parallel arrangement can be observed. As the cell cycle advances, the assembling of daughter cells (DC) can be observed, coordinated with the duplication of centrioles and the assembly of the mitotic spindle (ms) (panels II and III). In panel IV, two daughter cells inside the mother cells can be observed just before cytokinesis. Panels I and IV are Z-maxim intensity projections of the whole vacuole. Panels II and III are Z- maxim intensity projections of selected slices to show the data of interest. The scale bar is 1 µm for all images.

**Figure 4 microorganisms-12-00061-f004:**
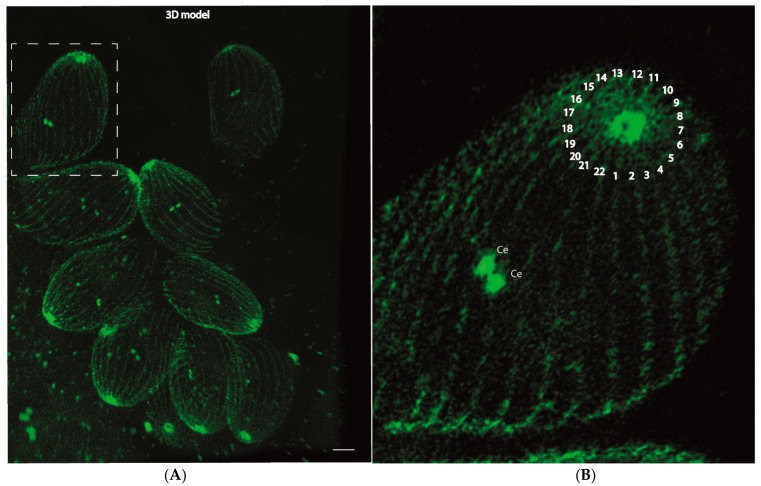
The two microtubule organizing centers of the cell of *Neospora caninum* under ultrastructure expansion microscopy. (**A**) Using a UExM image combined with AGAVE 3D v1.5.0 free software, we generated a 3D model of a vacuole of *N. caninum* stained with acetylated tubulin. (**B**) Using the 3D model shown in panel (**A**), we could observe the two MTOCs of this parasite, identifying and counting the number of subpellicular microtubules (labeled with the number 1–22). Ce stands for centrioles. The scale bar represents 1 µm.

**Figure 5 microorganisms-12-00061-f005:**
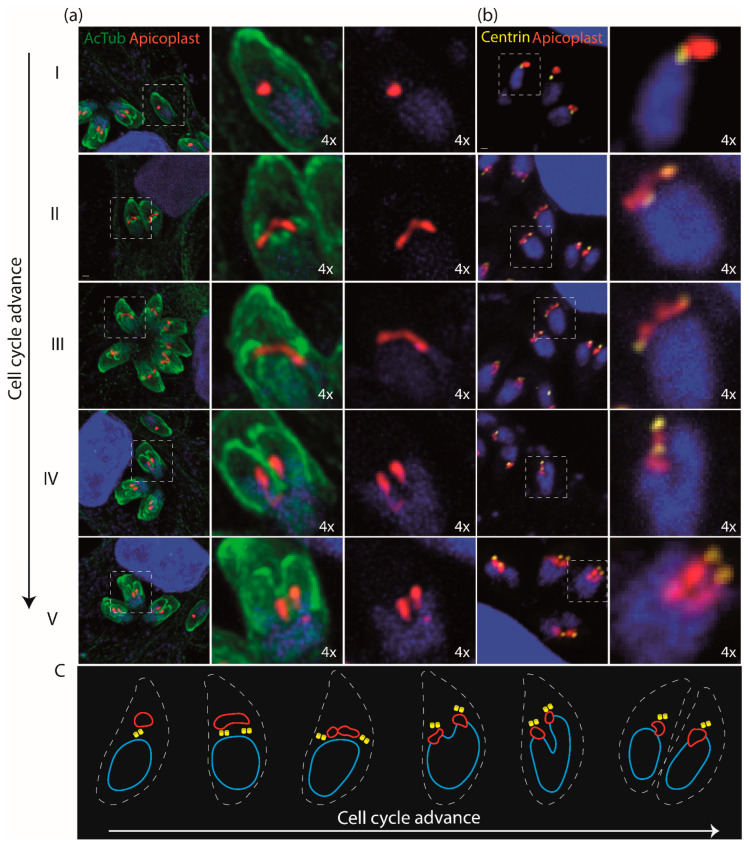
The apicoplast followed a cell-cycle-dependent dynamic in *Neospora caninum*. (**a**) IFA of *Neospora caninum* parasites stained with anti-cpn60 antibody (apicoplast-red), anti-acetylated tubulin (green) and DAPI (blue). Panels I–V indicate the different stages in cell cycle progression. Note that as the cell cycle advanced, the apicoplast elongated and distributed equitably to each daughter cell. (**b**) IFA of *Neospora caninum* parasites stained with anti-cpn60 antibody (apicoplast-red), anti-Centrin-1 (centrosome-yellow) and DAPI (blue). Note that as the cell cycle advanced, the apicoplast elongated and distributed, always attached to the centrosome. As the cell cycle advanced, it could be observed that the centrosome changed its relative position to the apicoplast. As previously described in *T. gondii* [30], five stages could be identified where the centrosome and the apicoplast had a characteristic relative position [32]. (**c**) Schematic representation of the centrosome and apicoplast relative position during the cell cycle of *Neospora caninum*.

## Data Availability

Data are contained within the article.

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
