# Peer review of "Insights into the Cell Division of Neospora caninum"

_microorganisms, 2023, doi:10.3390/microorganisms12010061_

Round 1

Reviewer 1 Report

Comments and Suggestions for Authors

The manuscript entitled ‘Insights into the cell division of Neospora caninum’ described the use of ultrastructure expansion microscopy (UExM) to evaluate centrosome and apicoplast dynamics during N. caninum tachyzoites replications. This is a novel and informative study, however, there are important issue that prevent it acceptance for publication in the current state. Authors do not provide any quantitative data to support many of their results. Stated results seems to come from single of limited number of observations, which is not enough for solid establishing conclusions:

§  Results 3.1. Figure 1 support authors conclusion that N. caninum PV is disorganised in relation to Toxoplasma gondii. It is not clear in which specific parameters is this conclusion based on. Is this related to the distance between tachyzoites? Different sizes among different PVs? Different orientation of tachizoites? How many PV have been observed for this conclusion? At what specific time after infection this conclusion has been done, early and/or late PVs? Does this ‘disorganisation’ occurs since the beginning of replication of at any specific point after this has started? Interestingly, Figure 5 shows a very nice coordination of tachyzoites replication. And importantly, how these have been compared with T. gondii PV? Authors have not performed infections with T. gondii, therefore I wonder which studies have been use to observe PVs for comparison with Neospora PVs.

§  Similarly, in the discussion authors’ stated: ‘Remarkably N. caninum tends to build disorganized vacuoles, high in number with different parasite orientation, while T. gondii’s vacuoles tend to be smaller and tidier.’ What is this conclusion based on? This information has not been provided in the results section. Which is the average size for a Neospora PV and for Toxoplasma PV?

§  Results 3.2 section does not provide any relevant information (as better resolution observations are reported in results section 3.3). Is this to provide evidence that non UExM microscopy does not provide enough resolution for centrioles? If so, this could be integrated with the next section of just removed. In Figure 2, the modelling does not resemble the microscopical observation, since parasite membrane markers have not been used. Authors could overlay images with bright field to properly positioning nucleous in this images. The images do not support the information described in the main text, i.e. duplication and repositioning of centrosomes.

§  Results 3.4, Figure 4 allowed the quantification of 22 subpellicular microtubules. Has this quantification done form a single tachyzoite? Or how many tachyzoites have been used to do this same microtubule quantification? Variations in numbers have been provided for other coccidian (Burrell at al., 2022)

Other important issues are related to the lack of information provided in the introduction regarding centrosome in Apicomplexa parasites, some information is after scattered along the manuscript, but a clear description of the topic in the introduction would be very helpful to follow the study. In addition to this, Discussion section is extremely poor, authors just summarise they own results without doing significant discussion with current literature. There are a significant number of published articles related to this study which has not been taken into consideration (e.g. Sun et al., 2022; Dos Santos Pacheco et al., 2020; Periz et al., 2017).

Authors need to report their results in past tense, there is a continue mixture with present tense (this is detailed below, but some might be missing). There are also many missing details in Material and Methods section (specified below).

Other minor comments:

- Line 17: specify technique in Abstract

- Line 20: authors claimed in the Abstract ‘we aim to inspire innovative strategies for disease management’ however, nothing is mentioned again on this regard.

- Use of part tense: Line 13 explored; Line 15: described; Line 17: explored; Line 18: described; Line 20: aimed; Line 114: built; Line 149: indicated; Line 184: ensused; Line 187: unfolded; Line 190: materialised; Line 236: advanced; Line 237: changed; Line 239: displayed; Line 289: was

- Line 37: specify ‘in the intestine of canids’

- Line 41: Toxoplasma should be in full as it is the first time being named

- Line 44: italics for N. caninum

- Lines 47: rhoptries instead rohphtries

- Line 49-50: N. caninum relies its pathogenicity on its efficiency in the way it replicates = N. caninum pathogenicity relies on its replicative efficiency

- Line 56: T. gondii since has already been named before

- Line 75-79: how often and how the sub-culturing of cells and tachyzoites is done in terms of numbers

- Line 81: does authors mean 100% confluence, how many cells is this? Were cell monolayers washed before fixation?

- Line 85-86: please indicate specific labelling for these antibodies in Materials and Methods

- Line 106: what two-fold involves here? 1:2 dilution only?

- Line 122: does author mean parasitophorous vacuole, or intracellular vacuoles of the parasite?

- Line 124: T. gondii since has already been named before

- Line 124: could authors briefly described what is seen in T. gondii?

- Figure 1: add size bars

- Figure 1 and line 130: NHS ester is not mentioned in Materials and Methods, could authors describe what is this?

- Figure 1A: indicate HCN and PV, and other structures also in first panel

- Figure 1B: is the green fluorescence an specific staining or only an editing highlight? Please indicate in legend

- Line 140: T. gondii since has already been named before

- Line 142: Plasmodium in italics

- Line 151: progression instead advance; remove ‘or not’

- Line 131 remove ‘as a centrosome approach’

- Figures 2: Legend is very confusing. Size bars need to be added. How can authors know mitosis is happening if Dapi staining is not showing division?

- Line 155-158: please rephrase, it is not clear what authors mean here

- Line 159: how can authors recognise these are 4 centrioles with the low level of resolution?

- Line 159-161: where is this description in the figure, is there any visual evidence for this?

- Line 161-164: please rephrase, it is not clear and difficult to follow

- Figure 3: scale bars are missing

- Line 210-211: just UExM since has already been named in full before

- Figure 4: first panel is not a model, it is the actual microscopical observation. Panel 2 is missing the scale bar

- Line 246: When does apicoplast is repositioned again above the centrioles?

- Figure 5: Authors have to indicate in the legend what I-V means. Use past tense in the legend description.

- Line 260: please rephrase, ‘Apicomplexan relies on its pathogenicity’ is not accurate

- Line 263: ‘super tidy’ is not a scientific term

- Line 270: apicoplast and centrosome dynamics in N. caninum

- Line 276: genes, no proteins, are conserved. Is there any confirmation of these being expressed as proteins?

- Line 301: what do authors want to say (scientifically) with ‘tider’?

- Line 316: Data availability: authors have not modified the default paragraph

Comments on the Quality of English Language

Comments included withing the main review

Reviewer 2 Report

Comments and Suggestions for Authors

Neosporosis is a parasitic disease caused by the protozoan Neosporum caninum. It is a recently identified protozoan of the coccidia group. First described in Norway. Belongs to the phylum - Apicomplexa, class - Sporozoans, subclass - Coccidia, genus - NeosporaNeosporum caninum is an intracellular parasite that affects many animal organs. The disease is caused by tissue necrosis, which occurs when cysts rupture and invasion by tachyzoites. Their permanent hosts are dogs, wolves, and coyotes. Intermediate hosts – alpaca, antelope, sika deer, black-tailed deer, cattle, dogs (as a permanent and intermediate host), lyre deer, equines, goats, sheep, llamas, pine marten, rhinoceros, etc.

This topic is very important and interesting. The authors presented their work in great detail and in a high scientific style. The work contains five figures that clearly show and prove the morphology and physiology of Neosporum caninum

But I have to make a note. It is necessary to supplement the Scientific classification information: from Phylum to Species. After adding this information, the article can be accepted for publication.

Round 2

Reviewer 1 Report

Comments and Suggestions for Authors

The manuscript quality has been significantly improved after the authors has addressed major and minor issues identified in the previous version. Authors’ claims are now more in accordance with their observations, and aims for each results section has been clarified. Most minor comments has been amended and there are just a couple of minor comments to consider before publication: 

- Line 122-123 (previously 106): Authors have been amended this, but I suggest that they provide the specific dilution/concentration used. I do not agree with the concept ‘regular immunofluorescence assays’ since there are significant variations in concentration for every assay and laboratory, therefore it is not clear which are the concentrations in a ‘regular assay’. 

- Line 144: T. gondii instead Toxoplasma gondii

Author Response

We agree with the comments. We now modified the manuscript according to the suggestions.

In Lines 122-123, the new text (and paragraph) is:

"To visualize the ultrastructure of the parasites and discern the spatial distribution of specific proteins, a conventional indirect immunofluorescence protocol, as described previously for T. gondii, was applied [21]. Mouse anti-acetylated tubulin (acetylated tubulin structure-marker) (Sigma T7451) and goat anti-mouse Alexa Fluor 594 (Invitrogen) were both used in a dilution of 1:500 in PBS. "

In Lines 144 we wrote: T. gondii.

In addition, we added several changes related to English revision.

We thank the work of the reviewer.

Sincerely,

Carlos Robello
